# Molecular Structures and Spectral Properties of Natural Indigo and Indirubin: Experimental and DFT Studies

**DOI:** 10.3390/molecules24213831

**Published:** 2019-10-24

**Authors:** Zixin Ju, Jie Sun, Yanping Liu

**Affiliations:** 1Key Laboratory of Textile Science & Technology of Ministry of Education, College of Textiles, Donghua University, Shanghai 201620, China; irisju@126.com; 2Shanghai Naturalism Biological Technology Co., Ltd., Shanghai 201616, China; simon@teamwisdom.com.cn

**Keywords:** natural indigo, natural indirubin, FTIR, Raman, UV-Visible, fluorescence, DFT

## Abstract

This paper presents a comparative study on natural indigo and indirubin in terms of molecular structures and spectral properties by using both computational and experimental methods. The spectral properties were analyzed with Fourier transform infrared (FTIR), Raman, UV-Visible, and fluorescence techniques. The density functional theory (DFT) method with B3LYP using 6-311G(d,p) basis set was utilized to obtain their optimized geometric structures and calculate the molecular electrostatic potential, frontier molecular orbitals, FTIR, and Raman spectra. The single-excitation configuration interaction (CIS), time-dependent density functional theory (TD-DFT), and polarization continuum model (PCM) were used to optimize the excited state structure and calculate the UV-Visible absorption and fluorescence spectra of the two molecules at B3LYP/6-311G(d,p) level. The results showed that all computational spectra agreed well with the experimental results. It was found that the same vibrational mode presents a lower frequency in indigo than that in indirubin. The frontier molecular orbital analysis demonstrated that the UV-Visible absorption and fluorescence bands of indigo and indirubin are mainly derived from π → π* transition. The results also implied that the indigo molecule is more conjugated and planar than indirubin, thereby exhibiting a longer maximum absorption wavelength and stronger fluorescence peak.

## 1. Introduction

Natural indigo (C.I. 75780), mazarine powder extracted from the indigo plant, is one of the earliest and most popular vat dyestuffs [1,2]. Recently, the demand for natural indigo dye is increasing dramatically for its safety and biodegradability. Natural indigo is now being used by several fashion retail giants such as H&M on a large scale to replace synthetic indigo dye for developing safe, green, and sustainable jeans. Since natural indigo originates from natural resources, it is treble the cost of synthetic indigo dye. Clear evidence of dyeing fabrics with natural indigo is needed to prevent adulteration and protect the interests of consumers [3]. Natural indigo is easily oxidized to indirubin during the extraction process, so the presence of indirubin and other unknown impurities is a key feature of natural indigo dye. In this regard, an in-depth understanding of similarities and differences in the molecular structures features and various vibrational spectra of indigo and indirubin molecules is critical for establishing the authenticity of natural indigo dye.

Indirubin (C.I. 75790), an isomer of indigo, which is an effective component of anti-leukemia agent, also has pharmacological effects such as sterilization. The difference in molecular structures of indigo and indirubin gives rise to different physical and chemical properties as well as their applications. As shown in Figure 1, indigo and indirubin with the molecular formula of C_16_H_10_N_2_O_2_ have a molecular weight of 262.6 Daltons, both of which have the double indole structure. Indigo is a planar molecule with a symmetrical trans structure and a strong conjugation effect [4], in which two intramolecular hydrogen bonds are formed between the adjacent carbonyl groups and the imino groups. Indirubin is an asymmetric structural isomer of indigo by condensing two indole rings [5]. In their production processes, the conditions of reaction should be strictly controlled in order to avoid interconversion of indigo and indirubin [6].

Previous works on indigo mainly focused on the dyeing process optimization and ancient textile identification [7,8,9]. In order to improve the dyeing performance of indigo and expand its application in electronics and other fields, considerable research efforts have been devoted to the DFT studies of indigo and its derivatives [10,11,12,13,14]. The earliest investigation on vibrational spectra of indigo has been conducted at the HF/3-21G level [15,16]. Later on, recalculations were carried out with more accurate methods or at larger basis sets, such as B3LYP/6-31G(d,p) [11], B3LYP/LANL2DZ/6-31G(d,p) [17], and B3LYP/6-311G(d,p) [18]. A recent study confirmed the assignments of the main experimental features of the FTIR and Raman spectra below 1700 cm^−1^ by using B3LYP/6-311++G(d,p) [19]. Theoretical studies on the differences between the absorption of indigo molecule and its derivatives have been reported at TD-DFT/B3LYP/6-31G(d) [12], and their differences in emission spectra have also been investigated at the TD-DFT/B3LYP/6-31G(d,p) level [13]. The theoretical results indicated that the lowest excited singlet states of indigo and its derivatives were derived from the HOMO-LUMO transition. In addition to the studies based on a small basis set that neglected the solvent effect [12,13], the effects of environmental solvents and substitution patterns were assessed at TD-DFT/B3LYP/6-311+G(2d,p) [14]. It was found that indigo showed a strong bathochromic shift as the dielectric constant of the solvent increases due to the improved stability of the charged separation structure. The fluorescence quantum yield of polymeric indigo is an order of magnitude lower than that of indigo as a result of the energy transfer between different indigo chromophoric units [20].

The studies on indirubin were mainly related to the medical field, such as drug metabolism process and anti-tumor mechanism. In order to improve the efficacy and safety of indirubin, the influence of substituents on the geometry configuration and anti-cancer activity of indirubin has been studied computationally [21,22,23,24]. The optimized geometric structure and main vibrational modes assignment of indirubin were investigated at B3LYP/6-31G(d) level, but the vibrational spectra were not completely interpreted [22]. The frontier orbital study showed that the electrons in indirubin can jump easily due to the low HOMO-LUMO energy gap [23]. Computational study for the absorption spectra of indirubin by using hybrid methods B3LYP and PBE0 showed that the excitation responsible for the color typically corresponds to the HOMO → LUMO transition, which is a typical π → π* transition with densities originating from the central double bond to the side single bonds [24]. 

Some efforts have already been made to investigate indigo, indirubin, and their derivatives computationally. However, the assignments of the whole vibrational spectral range of indirubin are still lacking. The excited state characterization of indirubin also has not been reported. To establish the authenticity of natural indigo dye, this work compared the molecular structural differences between indigo and indirubin from natural resources, comprehensively characterized their spectral properties, and analyzed the relationship between the luminescent mechanism, spectral characteristics, and molecular structures of the two molecules. Accordingly, the FTIR, Raman, UV-Visible, and fluorescence spectral properties of indigo and indirubin have been measured and calculated. It is expected that this study could provide a reference for the spectral analysis of indigo and indirubin and lay the scientific basis for the assessment of purity and identification of natural indigo dye.

## 2. Results and Discussion

### 2.1. Geometry Optimization

The computational optimized geometries of indigo and indirubin at B3LYP/6-311G(d,p) basis set are shown in Figure 2. The theoretical optimized structural parameters (bond distances and bond angles) were compared with the experimental data in the literature [4,25], and the results are listed in Table 1. The correlation coefficient *R^2^* for bond distances and angles of indigo are 0.95398 and 0.96445, respectively. The corresponding values for bond distances and angles of indirubin amount to 0.8891 and 0.9660. Exemplary correlations are presented in Appendix A. The slight deviations between the experimental and computational results can be attributed to the different phases, the computational system is only a single molecule, while the latter is a solid phase with inter-molecular interactions affect.

The data in Table 1 indicate that the C–C bond lengths of the benzene rings in indigo and indirubin are, respectively, ranging from 1.386 to 1.412 Å and from 1.360 to 1.430 Å, which are close to the C–C bond length (1.4 Å) of the benzene molecule, showing a conjugation effect. The C–C bond lengths of the heterocyclic rings in indigo and indirubin are 1.424-1.495Å and 1.456–1.530 Å, respectively. The two bond length ranges are just in between those of the C–C single bond (1.540 Å) and the C=C double bond (1.340 Å), so the C–C bonds were involved in the overall conjugation of the two molecules. The N(11)–C(13) and N(26)–C(28) bond lengths of the heterocyclic ring in indigo are identical being 1.379 Å (1.382 Å in Exp.). The N(11)–C(13) and N(26)–C(27) bond lengths of the heterocyclic ring in indirubin are respectively 1.374 Å and 1.377 Å (1.40 Å and 1.38 Å in Exp.), which are within the lengths of the normal C–N single bond (1.480 Å) and C=N double bond (1.270 Å). Consequently, the C–N bonds in the two molecules are both delocalized bonds, which also participate in the overall conjugation of molecules.

Although there are many delocalized bonds both in the indigo and indirubin molecules, their conjugated degrees are different from each other. Different from the planar molecular indigo, the study on the X-ray crystal structure of indirubin has been reported that presumably due to the stereochemical constraints, and there is a small deviation (4°) from planarity between the two indole rings in indirubin [5,25]. This indicates that there is only one intramolecular hydrogen bond in indirubin between the N–H and C=O group while two in indigo. Therefore, the conjugation effect of the indirubin molecule is relatively weaker than that of the indigo molecule, and indigo is more conjugated and planar than indirubin. In addition, indigo is a nonpolar molecule with central symmetry and belongs to the C_2h_ point group. On the contrary, indirubin is a polar molecule and belongs to the C_s_ point group.

### 2.2. Analysis of Molecular Electrostatic Potential

The molecular electrostatic potential surface (ESP) implying the interaction between molecules presents the distribution of electrostatic potential (electron + nucleus). The color in the ESP surface is a measure of the electrostatic potential value. Red represents electron-rich and partially negative charge, blue represents electron deficient and partially positive charge, light blue represents slightly electron deficient region, yellow represents slightly electron rich region, and green represents neutral.

Figure 3 shows the total electron density mapped with the electrostatic potential surface of indigo (a) and indirubin (b). The regions around the two oxygen atoms linking with the carbon atom by the double bond both in indigo and indirubin represent the most negative potential region (red), whereas the two hydrogen atoms linking with the nitrogen atoms have the largest positive charge (blue). From the ESP surface, it is confirmed that the negatively charged oxygen atoms tend to form hydrogen bonds with the positively charged hydrogen in indigo and indirubin.

### 2.3. Frontier Molecular Orbitals

The frontier orbital theory holds that the energy gap between the highest occupied molecular orbital (HOMO) and the lowest unoccupied molecular orbital (LUMO) reflects the ability of electrons to transit from orbital to empty orbital, which to some extent represents the chemical activity of molecules. A molecule with a bigger energy gap is more inert and the electron transition is more difficult. 

The energy values of HOMO, LUMO, HOMO−1, LUMO+1 molecular orbitals, and the corresponding HOMO-LUMO energy gap of indigo and indirubin were computed at the theoretical level of B3LYP/6-311G(d,p). The wave function Multiwfn 3.6 combined with the visualization software VMD 1.9.3 was used to draw the frontier molecular orbits of indigo and indirubin as shown in Figure 4 [26,27]. As shown in Table 2, the HOMO-LUMO energy gap values of indigo and indirubin are 2.5035 eV and 2.7511 eV, respectively. In this connection, indirubin of a higher HOMO-LUMO energy gap possesses better chemical stability.

### 2.4. Vibrational Analysis

Vibrational spectra including FTIR and Raman are valuable for elaborating the structures and compositional characteristics of materials. In order to interpret experimental spectra accurately, it is of great importance to predict the wavenumbers and spectral intensities of the molecule and identify vibration modes with quantum chemical computations [28]. The experimental FTIR and Raman spectra of indigo and indirubin are presented in Figure 5a and Figure 5b, respectively. The Raman spectra were obtained only in the range of 100–1800 cm^−1^ due to the interference of their own fluorescence. Both experimental and computational vibrational wavelengths and assignments for Indigo and Indirubin at B3LYP/6-311G(d,p) basis set are detailed in Table 3, in which the vibrational modes of indigo and indirubin were identified by using Gauss View 6. There is no virtual frequency in the computational results indicates that the optimized configuration is indeed a minimum point on the potential energy surface and the compound can exist stably. All computational wavelengths had been scaled by 0.967 to correct the theoretical error in this work.

As can be seen from Table 3, slight discrepancies between the measured and simulated wavenumbers are attributed to the computational system that only a single molecule in the gas phase was adopted by neglecting inter-molecular interactions. There are 30 atoms and 84 fundamental vibrations in both indigo and indirubin molecules. Indigo belongs to the C_2h_ group with a center of symmetry, which holds the mutual exclusion rule that only the *gerade* vibrations are Raman active and the *ungerade* vibrations are IR active. On the contrary, there is no mutual exclusion rule in the polar indirubin molecule. 

The experimental vibrational spectra and assignments of indigo are presented in Figure 5 and Table 3. The broad band observed at 3436 cm^−1^ in the FTIR spectrum is assigned to the stretching vibration of the N–H bond. The strongest intensity in IR band located at 3269 cm^−1^ is assigned to the stretching vibration of the chelation structure of C=O and hydrogen bonds. The stretching vibration of C–H in the aromatic ring is between 2853 and 3060 cm^−1^ in the IR spectrum. The spectral range of 1586–1701 cm^−1^ is assigned to the stretching vibration of the conjugated system of C=C, C=O, and N–H groups, and this range is in response to indigoid molecules [19,29]. The ring C–C stretching bands also appeared in the FTIR and Raman spectrum at 1483 and 1462 cm^−1^. The region below 1394 cm^−1^ can be mainly ascribed to the deformation vibrations of N–H and C–H.

The experimental vibrational spectra and assignments of indirubin are also given in Figure 5 and Table 3. The strongest intensity IR band located at 3433 cm^−1^ is assigned to the stretching vibration of the N–H bond. The stretching vibration of the chelation structure of C=O and hydrogen bond appears at 3346 cm^−1^ in the FTIR spectrum. The spectral range of 3230–2853 cm^−1^ is corresponding to the stretching vibration of C–H in the aromatic ring. The stretching vibration of the conjugated system of C=C, C=O, and N–H groups is observed between 1587 and 1703 cm^−1^. The ring C–C stretching bands are also observed at 1482 and 1465 cm^−1^ in IR (1479 and 1460 cm^−1^ in Raman). The deformation vibrations of N–H and C–H can be assigned below 1385 cm^−1^. 

The assignments show that the corresponding vibrational modes in indigo and indirubin present different wavenumbers and Raman shifts, indicating their different symmetries. The strongest intensities of indigo are at 3269 cm^−1^ (FTIR) and at 1572 cm^−1^ (Raman), while those of indirubin are located at 3433 cm^−1^ and 1587 cm^−1^, respectively. The same vibrational mode presents a higher frequency in indirubin than that in indigo because indirubin is more polar and less conjugated. The intermolecular hydrogen bonds in indigo and indirubin are two and one, respectively, so the IR band located at 3269 cm^−1^ in indigo is wider than the IR band located at 3346 cm^−1^ in indirubin, because of stretching vibration of chelation structure of C=O and hydrogen bonds.

### 2.5. UV-Visible Spectra Analysis

The UV-Visible absorption spectra of indigo and indirubin obtained in the experiments are plotted in Figure 6. Both experimental and computational absorption wavelengths, transition coefficient, oscillator strengths, and assignments for indigo and indirubin are detailed in Table 4.

As can be seen from Table 4 that indigo has two characteristic absorption peaks at 268 and 287 nm in the ultraviolet region and one characteristic absorption peak at 612 nm in the visible light region. Indirubin has two characteristic absorption peaks at 293 and 363 nm in the ultraviolet region and one characteristic absorption peak at 547 nm in the visible light region. The maximum absorption wavelengths of indigo and indirubin are 612 and 547 nm, respectively, both of which are higher than the computational values. This may be due to neglecting intermolecular interactions between molecules and solvents in the calculations. Indigo and indirubin molecules are large conjugated systems, which results in a broad absorption peak at 612 and 547 nm, and the maximum absorption wavelength generated a bathochromic shift. As reported before, the maximum absorption at 612 and 547 nm are assigned to the transition from HOMO to LUMO orbital, which has a π → π* character [29]. Due to the smaller degree of conjugation of indirubin, its maximum absorption wavelength is shorter. The strongest absorption bands at 268 nm (indigo) and 293 nm (indirubin) are signed to HOMO−1 → LUMO+1 transition that also has a π → π* feature. This difference might be because the positions of the carbonyl and the imino group in the two molecules are different, resulting in different degrees of steric hindrance effect in the molecules. Consequently, the planar properties, conjugated degree, and absorption wavelength of the two molecules are different.

### 2.6. Fluorescence Spectra Analysis

The electrons in the π* orbit are unstable and quickly return to the ground state, thereby generating fluorescence. The first singlet excited state generated a π → π* transition when absorbing energy, because of their large degree of conjugation. The fluorescence spectra of indigo and indirubin were obtained in the experiments as shown in Figure 7. Experimental and computational emission wavelengths, transition coefficient, oscillator strengths, and assignments for indigo and indirubin are detailed in Table 5.

The experimental results in Table 5 show that the strongest fluorescence peak wavelengths of indigo and indirubin are 486 and 412 nm, respectively, both of which are mainly generated by excitation from LUMO → HOMO. The difference in peak wavelength is ascribed to that indigo molecules in the excited state have better planar conformation than indirubin which can reduce the molecular vibration, so that the excitation energy of the molecules is not easily released by thermal energy due to vibration. This is beneficial to the generation of fluorescence. Additionally, indigo molecules have a large degree of conjugation, so its fluorescence wavelength is longer, and the peak intensity is also significantly higher than those of indirubin. The computational values agree well with the experimental results, proving that the ground and excited state configuration of the optimized molecule are acceptable.

## 3. Experimental

### 3.1. Materials

Indigo (98%) and indirubin (97.5%) extracted from Folium Isatidis were purchased from Chengdu PureChem-Standard Co., Ltd. (Chengdu, China) and used as received. The solvent used, N,N-dimethylformamide (DMF, 99.5%) of analytical grade, was obtained from Sinopharm Chemical Reagent Co., Ltd. (Shanghai, China). Two 100 mL volumetric flasks with 5 mg indigo and 5 mg indirubin were prepared, and DMF was added to the marks of the two flasks, tightly covered with parafilm, and placed in an ultrasonic bath for 30 min. The solutions of indigo and indirubin of 100 μg ml^−1^ in DMF were obtained.

### 3.2. Instrumentation

#### 3.2.1. Infrared Spectroscopy

The room temperature FTIR spectra of indigo and indirubin in the frequency ranging from 4000 to 400 cm^−1^ were recorded on a Fourier Infrared Raman spectrometer (FTIR, NEXUS-670, Nicolet, Madison, WI, USA) by using a potassium bromide tableting method. The background and samples were scanned 16 times with a resolution of 4 cm^−1^. 

#### 3.2.2. Raman Spectroscopy

The Raman spectra of indigo and indirubin at room temperature were measured in the 1800 and 100 cm^−1^ intervals with an InVia Reflex Micro Raman Spectrometer (Renishaw, Gloucester, UK). The Raman data were collected by using the Ar laser of 532 nm and a 50× objective.

#### 3.2.3. UV–Visible Spectroscopy

UV-visible spectra measurements were recorded on a Lambda 35 UV–Visible NIR spectrophotometer (PerkinElmer, Waltham, MA, USA) using a quartz cuvette. The solutions of indigo and indirubin in DMF were prepared for this test. The spectra were recorded between 265 and 750 nm.

#### 3.2.4. Fluorescence Spectrometry

Fluorescence spectra were recorded using a quartz cuvette on the QM/TM steady state/transient fluorescence spectrometer (PTI, Birmingham, NJ, USA). All slit widths were set to 5 nm, and the emission wavelength of 480 nm was selected for the 100 μg ml^−1^ indigo/DMF solution. The excitation spectrum was recorded between 260 and 460 nm, from which the optimal excitation wavelength was selected. The emission spectrum between 400 nm and 680 nm was recorded at 380 nm. Under the same conditions, the excitation spectrum of the 100 μg ml^−1^ indirubin/DMF solution was measured, and the optimum excitation wavelength was set as 350 nm. Its emission spectrum between 370 nm and 680 nm was recorded.

### 3.3. Computational Details

With the use of the density functional theory (DFT) method, the optimized geometric structures, vibrational frequency, molecular electrostatic potential, and frontier molecular orbitals of indigo and indirubin were calculated at Becke3-Lee-Yang-Parr (B3LYP) with 6-311G(d,p) basis set. The single-excitation configuration interaction (CIS) was used to optimize the excited state structure. Based on the optimized structure, the time-dependent density functional theory (TD-DFT) and the polarization continuum model (PCM) were used together to calculate the absorption spectra of the two molecules in the ground state configuration and the fluorescence spectra in the excited state configuration at B3LYP with 6-311G(d,p). All the calculations were performed using the program package Gaussian 16 [30]. Gauss view 6 was used to build Gaussian input files and visualize the results of the Gaussian calculations [31].

## 4. Conclusions

The molecular structures and spectral properties of indigo and indirubin from natural resources have been characterized experimentally and computationally. The steric hindrance effect, plane and conjugated degree of the two molecules were different, giving rise to different spectral properties. The main conclusions can be drawn as below.

(1) It is confirmed that both indigo and indirubin molecules form a large stable conjugated system. Indirubin has a higher frontier orbital energy gap and better chemical stability than indigo.

(2) The strongest intensities of indigo are at 3269 cm^−1^ (FTIR) and at 1572 cm^−1^ (Raman), but those of indirubin are located at 3433 cm^−1^ and 1587 cm^−1^, respectively.

(3) The main absorption bands of indigo and indirubin are mainly assigned to the transition from HOMO → LUMO orbital and HOMO-1 → LUMO+1 orbital, which have a π → π* character. Indirubin has lower planeness, smaller degree of conjugation than indigo, thereby having smaller maximum absorption wavelength.

(4) The fluorescence wavelength of indigo is longer, and the peak intensity is also significantly higher than those of indirubin. The strongest fluorescence peak wavelengths of indigo and indirubin are mainly generated by excitation from LUMO → HOMO, which have a π → π* character. Indigo molecules in the excited state have better planar properties and conjugation, which can reduce molecular vibrations, so the excitation energy of the molecules is not easily released by thermal energy due to vibration. This is the reason why indigo generates strong fluorescence.

The vibrational spectra of indigo and indirubin are different in various wavelengths, so synthetic indigo without any impurities could have different features compared with natural indigo containing the main impurity, indirubin. This provides a practicable way to distinguish natural and synthetic indigo by using vibrational spectra.

## Figures and Tables

**Figure 1 molecules-24-03831-f001:**
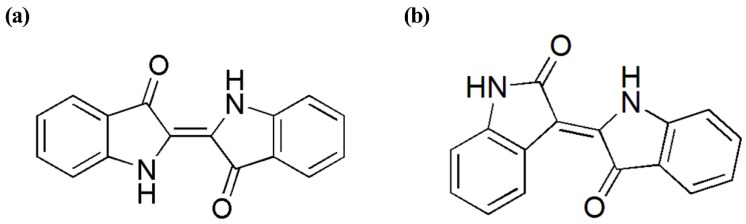
Chemical structures of indigo (**a**) and indirubin (**b**).

**Figure 2 molecules-24-03831-f002:**
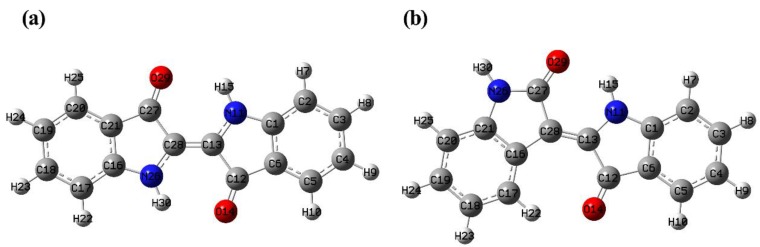
Optimized molecular structures of indigo (**a**) and indirubin (**b**) at B3LYP/6-311G(d,p).

**Figure 3 molecules-24-03831-f003:**
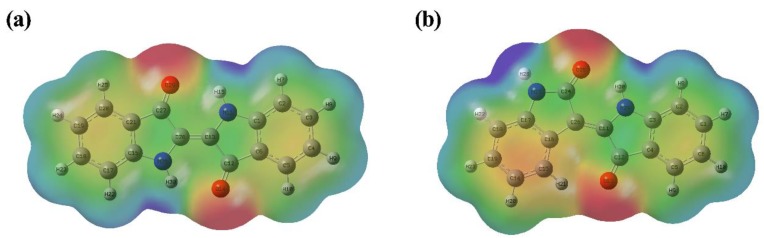
Total electron density mapped with electrostatic potential surface of indigo (**a**) and indirubin (**b**).

**Figure 4 molecules-24-03831-f004:**
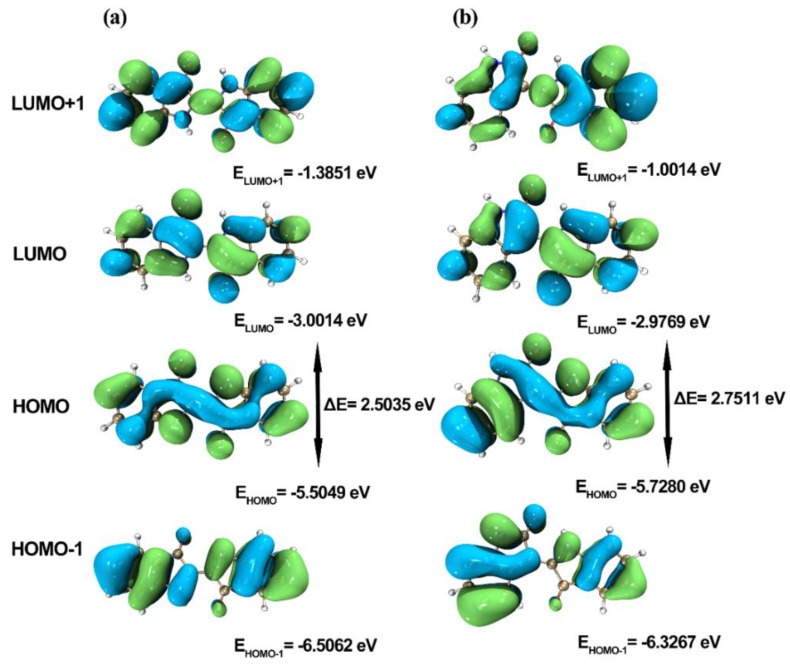
Frontier molecular orbitals of indigo (**a**) and indirubin (**b**).

**Figure 5 molecules-24-03831-f005:**
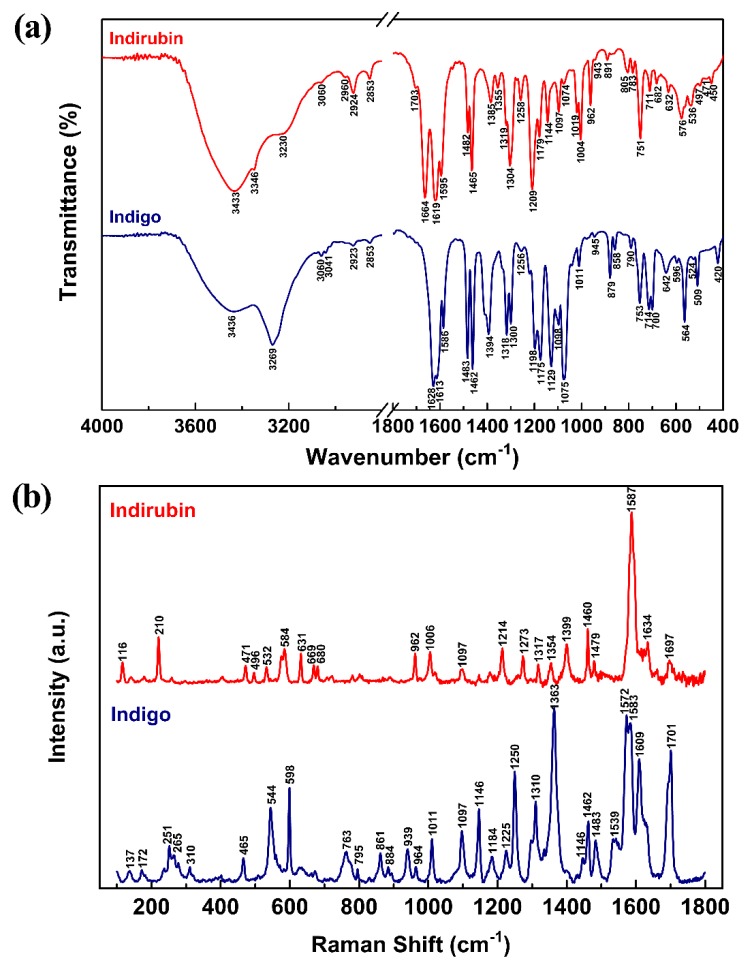
Infrared spectra (**a**) and Raman spectra (**b**) of indigo and indirubin.

**Figure 6 molecules-24-03831-f006:**
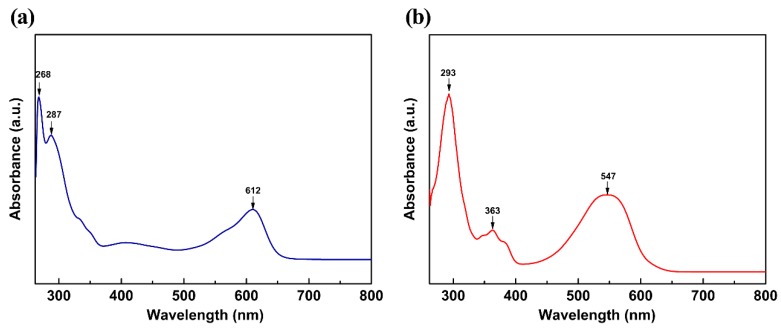
UV-Visible absorption spectra of indigo (**a**) and indirubin (**b**).

**Figure 7 molecules-24-03831-f007:**
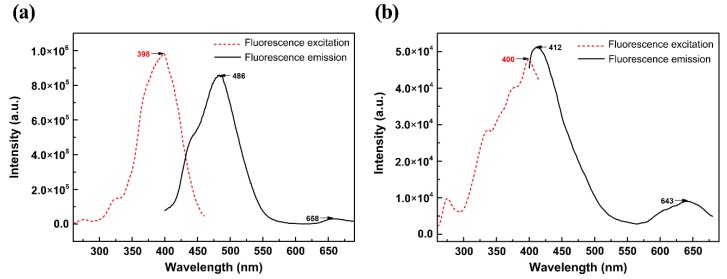
Fluorescence spectra of indigo (**a**) and indirubin (**b**).

**Table 1 molecules-24-03831-t001:** Optimized structural parameters for indigo and indirubin at B3LYP/6-311G(d,p).

Structural Parameters	Indigo		Structural Parameters	Indirubin	
Internuclear Distance (Å)	Calc.	Exp. [5]	Internuclear Distance (Å)	Calc.	Exp. [25]
C(1)-C(2)	1.393	1.393	C(1)-C(2)	1.391	1.43
C(2)-C(3)	1.394	1.387	C(2)-C(3)	1.396	1.41
C(3)-C(4)	1.403	1.386	C(3)-C(4)	1.401	1.41
C(4)-C(5)	1.390	1.392	C(4)-C(5)	1.392	1.41
C(5)-C(6)	1.393	1.387	C(5)-C(6)	1.391	1.38
C(1)-C(6)	1.412	1.404	C(1)-C(6)	1.405	1.39
C(16)-C(17)	1.393	1.393	C(16)-C(17)	1.397	1.41
C(17)-C(18)	1.394	1.387	C(17)-C(18)	1.396	1.42
C(18)-C(19)	1.403	1.386	C(18)-C(19)	1.395	1.41
C(19)-C(20)	1.390	1.392	C(19)-C(20)	1.389	1.41
C(20)-C(21)	1.393	1.387	C(20)-C(21)	1.384	1.36
C(21)-C(16)	1.412	1.404	C(21)-C(16)	1.417	1.40
C(6)-C(12)	1.468	1.424	C(6)-C(12)	1.471	1.48
C(12)-C(13)	1.493	1.495	C(12)-C(13)	1.524	1.53
C(21)-C(27)	1.468	1.424	C(16)-C(28)	1.456	1.51
C(27)-C(28)	1.493	1.495	C(27)-C(28)	1.496	1.48
N(11)-C(13)	1.379	1.382	N(11)-C(13)	1.374	1.40
N(26)-C(28)	1.379	1.382	N(26)-C(27)	1.377	1.38
C(1)-N(11)	1.387	1.380	C(1)-N(11)	1.389	1.40
C(12)-O(14)	1.226	1.240	C(12)-O(14)	1.216	1.21
C(13)-C(28)	1.358	1.342	C(13)-C(28)	1.368	1.31
C(27)-O(29)	1.226	1.240	C(27)-O(29)	1.226	1.25
Bond angle (°)	**Calc.**	**Exp. [5]**	Bond angle (°)	**Calc.**	**Exp. [25]**
C(1)-N(11)-C(13)	109.969	108.63	C(1)-N(11)-C(13)	111.654	110
C(6)-C(12)-O(14)	130.765	129.13	C(6)-C(12)-O(14)	128.603	128
C(1)-N(11)-H(15)	127.957	130.30	C(1)-N(11)-H(15)	128.015	--
C(1)-C(6)-C(12)	107.676	106.03	C(1)-C(6)-C(12)	107.884	106
C(6)-C(12)-C(13)	104.199	107.23	C(6)-C(12)-C(13)	104.402	106
C(12)-C(13)-N(11)	108.355	107.23	C(12)-C(13)-N(11)	106.487	106
C(6)-C(1)-N(11)	109.800	110.83	C(6)-C(1)-N(11)	109.573	112
C(16)-C(21)-C(27)	107.676	106.03	C(16)-C(21)-N(26)	109.065	110
C(21)-C(27)-C(28)	104.199	107.23	N(26)-C(27)-C(28)	105.995	108
C(27)-C(28)-N(26)	108.355	107.23	C(27)-C(28)-C(16)	106.622	105
C(16)-N(26)-C(28)	109.969	108.63	C(21)-N(26)-C(27)	111.544	111
C(21)-C(16)-N(26)	109.800	110.83	C(21)-C(16)-N(28)	106.774	107
N(11)-C(13)-C(28)	125.977	124.93	N(11)-C(13)-C(28)	123.244	122
C(21)-C(27)-O(29)	130.765	129.13	N(26)-C(27)-O(29)	125.564	126
C(16)-N(26)-H(30)	127.957	130.30	C(21)-N(26)-H(30)	126.055	--

**Table 2 molecules-24-03831-t002:** Total molecular energy and frontier orbital energy.

Thermodynamic Parameters (298 K)	Indigo	Indirubin
E_LUMO+1_ (eV)	−1.3851	−1.0014
E_LUMO_ (eV)	−3.0014	−2.9769
E_HOMO_ (eV)	−5.5049	−5.7280
E_HOMO−1_ (eV)	−6.5062	−6.3267
ΔE_LUMO−HOMO_ (eV)	2.5035	2.7511

**Table 3 molecules-24-03831-t003:** Experimental and computational vibrational wavelengths and assignments for indigo and indirubin at B3LYP/6-311G(d,p).

Mode	Indigo				Indirubin			
	Experimental		Scaled Computational	Tentative Assignment ^1^	Experimental		Scaled Computational	Tentative Assignment ^1^
	IR	Raman			IR	Raman		
1	--	--	3487.34	ν_s_ N–H	3433	--	3530.01	ν N–H
2	3436	--	3486.36	ν_as_ N–H	3346	--	3398.59	ν N–H
3	--	--	3091.68	ν_s_ C–H	3230	--	3124.20	ν C–H
4	3269	--	3091.54	ν_as_ C–H	3060	--	3091.65	ν C–H
5	3060	--	3082.39	ν_as_ C–H	2960	--	3084.44	ν C–H
6	3041	--	3082.38	ν C–H	2924	--	3083.39	ν C–H
7	--	--	3076.08	ν_s_ C–H	--	--	3076.65	ν C–H
8	2923	--	3076.07	ν_as_ C–H	2853	--	3073.11	ν C–H
9	2853	--	3063.62	ν_as_ C–H	--	--	3064.40	ν C–H
10	--	--	3063.60	ν_s_ C–H	--	--	3061.94	ν C–H
11	--	1701	1707.32	ν C=C + ν C=O	1703	--	1710.64	ν_s_ C=O + ν C=C
12	1628	--	1654.71	ν C=O + δ N–H	1664	1697	1686.27	ν_as_ C=O + δ N–H
13	--	1609	1626.18	ν C=C +ν C=O + ν C–C	--	--	1611.54	ν C=C + ν C–C + δ C–H
14	1613	--	1597.98	ν C–C_ring_ + δ C–H	1619	1634	1603.14	ν C–C + δ C–H
15	--	1583	1590.48	ν C=C +ν C=O + ν C–C	--	--	1582.29	ν C–C + δ N–H + δ C–H
16	--	1572	1572.14	ν C–C_ring_ + δ N–H	1595	1587	1578.62	ν C–C + ν C=C + δ N–H
17	1586	--	1571.73	ν C–C_ring_	--	--	1566.60	ν C–C + δ C–H
18	1483	--	1468.51	ν C–C + δ C–H + δ N–H	1482	1479	1466.72	ν C–C + δ C–H + ν N–H
19	--	1483	1467.51	ν C–C + δ C–H + δ N–H	--	--	1461.41	ν C–C + δ C–H + ν N–H
20	--	1462	1441.41	ν C–C_ring_ + δ C–H	1465	1460	1447.83	δ C–H + ν C–C
21	1462	--	1439.48	ν C–C_ring_ + δ C–H	--	--	1442.37	ν C–C + δ C–H
22	1394	--	1399.78	δ N–H + δ C–H	1385	1399	1390.31	δ N–H + δ C–H + ν C–C
23	--	1363	1361.05	δ N–H + δ C–H	1355	1354	1364.94	δ N–H + δ C–H + ν C-N
24	--		1340.83	ν C–C + ν C-N + δ C–H	--	--	1324.29	ν C–C + ν C-N + δ C–H
25	1318	--	1310.37	ν C–C_ring_ + δ C–H	--	1317	1308.36	ν C–C + δ C–H
26	--	1310	1299.01	δ C–H + ν C–C_ring_	1319	--	1276.29	δ C–H + ν C-N + ν C–C
27	1300	--	1276.19	δ C–H	1304	1273	1272.39	δ C–H + δ N–H + ν C–C
28	1256	--	1236.96	ν C-N +ν C–C + δ C–H	--	--	1255.96	δ C–H + ν C-N + ν C–C
29	--	1250	1226.74	ν C-N +ν C–C + δ C–H	--	--	1225.67	δ N–H + δ C–H + ν C-N
30	--	1225	1201.04	δ N–H + δ C–H	--	--	1198.25	δ N–H + δ C–H
31	1198	--	1169.72	ν C–C + δ C–H	1258	1214	1173.76	δ C=C + δ N–H + δ C–H
32	--	1184	1168.23	ν C–C + δ C–H	--	--	1167.49	δ C–H + δ N–H + ν C–C
33	1175	--	1151.86	δ C–H + δ N–H + ν C-N	1209	--	1163.41	δ C–H + δ N–H + ν C-N
34	--	1146	1134.32	δ C–H	--	--	1140.71	δ C–H + δ N–H
35	1129	--	1122.09	δ C–H + δ N–H + ν C-N	1179	--	1132.16	δ C–H + δ N–H
36	--	1097	1080.76	δ C–H	1144	--	1085.63	δ C–H + ν C–C_ring_
37	1098	--	1079.58	δ C–C + δ C–H	1097	--	1079.63	δ C–H + ν C–C_ring_
38	1075	--	1051.04	ν C–C + δ C–C + δ N–H	1074	1097	1012.68	δ C–H + ν C–C_ring_
39	--	1011	1002.59	δ C–H	1019	--	1003.28	δ C–H + ν C–C_ring_
40	1011	--	1000.51	ν C–C_ring_ + δ C–H	1004	1006	979.67	ν C–C + δ N–H
41	--	--	958.97	τ C–H	--	--	966.82	γ C–H
42	--	--	958.94	ω C–H	--	--	960.63	γ C–H
43	945	--	933.35	ω C–H	962	962	944.80	δ C–C + δ N–H + δ C–H
44	--	--	933.23	τ C–H	--	--	934.39	γ C–H
45	--	939	920.76	δ C–C + δ C–H	943	--	925.97	γ C–H
46	879	--	856.57	δ C–C + δ C-N + δ N–H	891	--	868.21	δ C–C + δ N–H + δ C–H
47	--	861	845.06	δ C–C + δ N–H + δ C-N	--	--	855.81	δ C–C + δ N–H + δ C–H
48	858	--	841.66	ω C–H	--	--	844.28	γ C–H
49	--	--	840.65	τ C–H	--	--	842.74	γ C–H
50	--	--	790.96	γ C–H + γ C–C	--	--	794.85	γ C–C + γ C–H
51	--	--	781.77	γ C–H + γ C–C	805	--	794.81	ν C–C + δ C–C + δ C–H
52	790	--	754.56	δ C–C + δ C–H	783	--	764.1	γ C=O + γ C-N + γ C–H
53	--	763	748.02	δ C=C + δ C-N	--	--	741.01	γ C–H
54	753	--	738.83	ω C–H	751	--	739.95	γ C–H + γ C–C_ring_
55	--	--	738.23	τ C–H	--	--	729.31	γ C–H + γ C–C_ring_
56	--	--	709.24	τ C–H	--	--	708.19	δ C–C + δ C–H + δ N-C
57	714	--	694.24	δ C–C + δ C-N	711	--	705.46	γ C–H + γ N–H + γ C–C
58	700	--	694.01	γ C–C + γ C–H	682	680	669.41	δ C–C + δ C–H + δ C-N
59	--	598	666.65	δ N–H + δ C–C + δ C–H	--	669	661.22	δ C–C + δ C–H + δ N–H
60	--	--	591.00	δ N–H + δ C–C + δ C–H	632	631	623.64	δ C–C + δ C–H + δ N–H
61	642	--	587.87	δ C–C +δ N–H +δ C=O	576	--	609.31	γ N–H
62	596	--	553.11	δ C–C + δ C–H	--	584	572.84	δ C–C + δ N–H + δ C=O
63	--	--	552.34	τ N–H + τ C–H	--	--	565.92	δ C–H + δ C=O + δ N–H
64	564	--	551.13	ω N–H +ω C–H	--	--	556.22	γ C–H + γ C–C
65	--	544	538.73	δ C–C + δ C–H	--	--	540.03	γ C–H + γ N–H + γ C–C
66	524	--	504.10	δ N–H +δ C–C	536	532	527.13	δ C–C + δ N–H + δ C–H
67	--	--	472.33	τ N–H	497	496	494.40	γ N–H
68	509	--	465.70	ω N–H	471	471	483.97	δ C–H + δ C–C + δ C-N
69	--	465	443.70	γ C=C +τ C–H + τ N–H	450	--	449.31	γ N–H + γ C–H + γ C–C
70	420	--	418.13	ω C–H	--	--	424.65	γ N–H + γ C–H + γ C–C
71	--	--	395.95	τ C–H	--	--	398.24	γ N–H + γ C–H + γ C–C
72	--	--	372.22	ω C-N + ω C–C	--	--	359.51	δ C=O + δ C–H + δ C–C
73	--	310	300.02	δ C–H + δ C=O + δ C–C	--	--	342.35	γ C–H + γ N–H + γ C–C
74	--	--	280.89	δ C–H + δ C=O + δ C–C	--	--	290.95	δ C=O + δ N–H + δ C–H
75	--	265	251.28	τ C-N + τ C–H + τ C–C	--	--	268.83	γ C-N + γ C–C + γ C–H
76	--	251	242.76	ρ C–H	--	210	250.91	δ C=O + δ C–C
77	--	--	223.97	ω C–C +ω C-N	--	--	235.48	γ C–C + γ C-N + γ C–H
78	--	172	222.05	δ C=O + δ C–H + δ N–H	--	--	215.76	δ C–H
79	--	--	155.98	ω C=O +ω C–H	--	--	144.67	γ C–H + γ N-C + γ C=O
80	--	137	153.90	τ C=O + τ C–H	--	116	139.47	γ C–H + γ N-C + γ C=O
81	--	--	93.86	τ C–H + τ ring	--	--	108.76	γ C–H + γ C=O
82	--	--	81.45	τ ring	--	--	103.18	δ C–H
83	--	--	67.99	δ C–H + δ N–H	--	--	61.04	τ ring
84	--	--	29.29	ω ring	--	--	38.09	ω ring

^1^ ν: stretching; ν_s_: symmetrical stretching; ν_as_: asymmetrical stretching; δ: in-plane deformation; γ: out-of-plane deformation; ρ: rocking; ω: wagging; τ: twisting.

**Table 4 molecules-24-03831-t004:** Theoretical UV-Visible absorption spectra of indigo and indirubin computational at B3LYP/6-311G(d,p) level of theory.

Molecular	Wavelength (nm)		Transition Coefficient	Oscillator Strengths (f)	Assignment ^1^
	Experimental	Computational			
Indigo	612	579.77	0.70811	0.3342	H → L
	268	272.14	0.66336	0.5277	H−1 → L+1
Indirubin	547	536.36	0.70152	0.2591	H → L
	293	259.85	0.52164	0.1432	H−1 → L+1

^1^ H = highest occupied molecular orbital; L = lowest unoccupied molecular orbital.

**Table 5 molecules-24-03831-t005:** Theoretical electronic emission spectra of indigo and indirubin computational at the B3LYP/6-311G(d,p) level of theory.

Molecular	Wavelength (nm)		Transition Coefficient	Oscillator Strengths (f)	Assignment ^1^
	Experimental	Computational			
Indigo	486 (8.57 × 10^5^)	438.35	0.68114	0.7352	L → H
Indirubin	412 (5.10 × 10^4^)	433.26	0.67598	0.6871	L → H

^1^ H = highest occupied molecular orbital; L = lowest unoccupied molecular orbital.

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
