# Peer review of "Molecular Structures and Spectral Properties of Natural Indigo and Indirubin: Experimental and DFT Studies"

_molecules, 2019, doi:10.3390/molecules24213831_

Round 1
Reviewer 1 Report
Review of manuscript "Molecular Structures and Spectral Properties of Natural Indigo and Indirubin: Experimental and DFT Studies"
Authors: Zixin Ju , Jie Sun , Yanping Liu
The manuscript is well designed and present the characterization of indigo and indirubin both experimentally and computationally. This is very well done and the results are well presented and compared. The various tables with the energies, assignments are well dome and will be useful for various researchers in various fields.
The application of such in-depth study have been laid out at the end of the conclusion but could be emphasized before in the manuscript, especially in the abstract or introduction (or maybe I just missed it, which also calls for pointing it out more).
I would advise for publication upon some minor changes, mostly in the form.
1) The labels in the figures needs to be bigger. The authors refer to bands in the text and it is very difficult to read the band wavelengths on the corresponding figures.
2) some minor grammatical errors should be addressed
line 191: I believe "more inert" is grammatically correct line 223-225: rephrase. The sentence is a bt confused to the reader line 237: "are also appeared" should be replaced with "also appeared" line 242: same remark, "is appeared" should be replaced by "appears" line 242: since we are referring to a specific spectrum, place "the" before "FTIR spectrum" line 249: replace "but" by "while" line 251: replace "is higher polar" by "is more polar" line 324, place a coma between "impurity" and "indirubin"
Author Response
Response to Reviewer 1 Comments

Reviewer 2 Report
Comments:
Molecular structure and spectral properties of natural indigo and indirubin: experimental and DFT studies.
Authors report the photophysical properties of indigo and indirubin molecules with the aid of experimental and DFT studies. My point of view, authors fail to compare the geometry of optimized molecule with experimental geometry. The SCXRD data for both compounds available in CCDC. The main objective of this manuscript is spectroscopy study on the well-known dye stuff. But author doesn’t explain unusual fluorescence behaviors of indigo. Particularly, authors fail to explain the experimental λem of indigo (λem: 486 nm) is shorter than the absorption (λmax: 612 nm), and λmax: 612 nm may be due to charge transfer. Author must be explaining how the structural modification influenced the absorption and emission of both compounds. I am not recommended in the current form of the manuscript to publish in Molecules.
For the above reason, I recommend publication of this work after major revision as follows:
1.- Both compounds are buying from the commercial sheller. so, author should be removed the claim and sentence, “indigo and indirubin extracted from Folium Isatidis” in abstract.
2.-Author must be compared with optimized geometry with experimental molecular geometry. This manuscript and Table 1, bond length is discussed in nm. It has any reason? Otherwise should be change to Å and its easy follow the readers. Some sentence in the section 3.1 Geometry optimization, line N°160-164 is not clear, and it should be correct it.
3.-Line N° 161. Author mentioned that two intramolecular interactions in indigo and one from indirubin. But indirubin also from two intramolecular interaction (N13-H15··O29=C27; and C17-H22···O14=C12), so should be motif it and the C17-H22···O14=C12 interaction is may be reason for difference in the conjugation and optical properties of both compounds and should be explain this.
4.-Please include unit of molecular weight in line N° 45.
5.-Author mentioned indirubin is a V shaped molecule in line N° 48 and my point of view is not a V shaped molecule. So should be remove it.
6.-Line N° 48, “two anthracene rings” is wrong it should be replaced by “two indole ring”
7.-Bond length of C13=C28 is 0.1358 nm in indigo and 0.1368 nm indirubin should be explain it. The C13=C28 bond is more elongate in indirubin than indigo. It means more conjugation observed in indirubin. Please check you sentence which related to the conjugation.
8.-Should be explain unusual fluorescence behaviors of indigo. Should be change the excitation wavelengths and study the fluorescence behaviors of both compounds.

Author Response
Response to Reviewer 2 Comments

Round 2
Reviewer 2 Report
The authors improved the present manuscript, so I approve it in the current form.